

# Edge computing task scheduling method based on user's social relations: a construction and solution for Smart City Library

Yun Teng[1] and Zijia Liu[2]

[1] Research Department, Jilin University of Finance and Economics, Changchun, Jilin, China
[2] Changchun Children's Library, Changchun, Jilin, China

## ABSTRACT

In the realm of the development of a Smart City Library, the integration of robust edge computing is vital. The research suggests a novel task-scheduling model for edge computing, leveraging user's social relationships. Analyzing these connections involves constructing a user's social relationship graph by implementing mathematical convolution and the Jaccard similarity ratio. This precise quantification of social ties ensures secure and reliable task scheduling. An equipment connection graph of a user equipment service is also crafted based on Euclidean distance, aligning task scheduling with device-to-device (D2D) communication conditions. Combining a user's social relationship graph and a user's device-service device connection graph creates a task-device bipartite graph. On the other hand, the calculation of a task execution cost and edge weight determination finalize a scheduling model. Implementing the proposed method for constructing a model for edge computing task scheduling based on utilizing the Kuhn–Munkres (KM) algorithm demonstrates positive impacts, which are few delays and less energy consumption, on edge computing task scheduling. For instance, when the social threshold score changes from 02. To 0.6, the total task execution delay time increases from 23 to 32, which is the best when compared with other algorithms. The approach strengthens security and reliability while decreasing task execution delays and energy consumption. This research advances edge computing for Smart City Libraries, promising transformative implications.

# INTRODUCTION

A large number of mobile devices and real-time applications often support the process of building a Smart City Library. The availability of advanced ICT tools such as smartphones and cloud computing has enabled libraries to improve their services and expand outreach. The article tries to conceptualize a methodology for a smart library information system that uses the smart attributes of recent technological means for library service improvements [36]. However, the growth of mobile devices and real-time applications has strained their

Corresponding author
Yun Teng, wintengyun@126.com

computing resources, resulting in a subpar user experience (*Yu et al., 2022*; *Goudarzi et al., 2020*). Edge computing addresses this issue by delegating computing tasks to edge devices like mobile phones and edge servers (*Fu et al., 2023*; *Wang et al., 2019*). However, user reluctance and device resource consumption challenge task offloading (*Lv et al., 2021*). An altruistic behavior in social networks offers task collaboration opportunities, and social relationships can be quantified for secure identification of mobile edge service devices. Trust information can then be implemented to schedule tasks through mobile device collaboration.

Device-to-device (D2D) technology is a communication technology that allows nearby devices to exchange information, easing data loading pressure on a core network (*Patsias et al., 2023*). A D2D connection between a mobile device acting as an edge server and resource-limited user equipment to aid task transmission can be constructed in mobile edge computing. A D2D communication is ideal for short-distance communications, offering higher link gain and transmission efficiency and contributing to battery conservation. A D2D has become a critical technology in the development of 5G (*Li G. Cai, 2020*).

Nevertheless, the prime objective in the investigation of the edge computing domain is the mitigation of challenges related to task execution delays and energy consumption. A pivotal hurdle arises in employing a user's mobile device as an edge server and task offloading through mobile device collaboration, driven by resource constraints and users' inherent self-interest.

The utilization of social relationships for task scheduling in edge computing has yet to be extensively researched. Nevertheless, social relationships between nodes are currently being researched in related fields such as the Internet of Things (IoTs) and fog computing. Recent studies have shown that effective and trusted collaboration can be achieved by employing a social relationship structure between mobile devices and wearable devices when users execute collaborative computing tasks. *Zhou et al. (2018)* proposed a collaborative mobile edge computing paradigm based on social motivation, employing the device social graph model to capture social relationships and implementing a two-part task offloading algorithm based on social perception (*Chen et al., 2018*). *Yang et al. (2015)* introduced a Top-K task offloading algorithm based on social relationships to improve the load balancing of task scheduling (*Yang et al., 2015*). *Yu et al. (2019)* suggested a socially conscious task offloading framework by combining a MEC offloading and a D2D offloading. In edge computing networks utilizing D2D communication, a data task scheduling based on a D2D device users' social relationships is also an important research direction. *Pu et al. (2016)* put forward a named data framework for social perception to implement D2D communication collaboration for content retrieval on edge devices, greatly saving cellular network traffic. *Yan et al. (2018)* suggested a cooperative perception data task forwarding mechanism for social D2D networks, which assessed user importance by employing differences in user interests, local centrality, and affinity to rationally select cooperative users and realize more efficient data task forwarding. However, the focus of the above approaches is to resolve a collaborative problem in a task scheduling process, and there has yet to be in-depth research and investigation on a task scheduling problem, which has limitations.

The research introduces a novel approach for formulating a task scheduling model in edge computing grounded in social relationships. The key contributions encompass (1) the examination and quantification of user social relationships through mathematical convolution and the Jaccard similarity ratio, leading to the creation of a comprehensive user social relationship graph; (2) the establishment of a user-device-service device connection graph, rooted in Euclidean distance principles to ensure compliance with D2D communication conditions; amalgamating a user's social relationship graph with this connection graph results in a task-device bipartite graph, where the execution costs of a computing task determine edge weights; (3) the implementation of a task scheduling model based on social relationships in edge computing, resolved through the utilization of the Kuhn–Munkres (KM) algorithm.

The methodology proposed for crafting task scheduling models based on social relationships in edge computing introduces several key innovations: (1) Leveraging D2D communication technology enhances transmission efficiency and contributes to the energy conservation of a battery. (2) Introducing quantification criteria rooted in investigating social relationships among users facilitates quantifying analysis results, and identifying secure and reliable mobile edge service devices. (3) The methodology collaboratively schedules tasks for mobile devices, enhancing user experience.

The overarching structure of the manuscript unfolds as follows: Section 'Introduction' provides an introduction, Section 'Related Work' delves into the related work, Section 'Construction of a Social Relationship-Based Model for Scheduling Edge Computing Tasks' establishes a model grounded in social relationships for scheduling edge computing tasks, Section 'Solution to Edge Computing Task Scheduling Model Using KM Algorithm' presents a KM algorithm-based solution for edge computing task scheduling models, Section 'Experimental Analysis' conducts a thorough performance assessment of the proposed method through experimental analysis, and Section 'Conclusion' concludes the research.

## RELATED WORK

Currently, many scholars have researched improving the performance of edge computing. Many computational tasks still need to be carried out at edge nodes due to insufficient computational performance of edge nodes and the transmission bandwidth of a WAN. Due to the limited computational power of edge nodes and transmission bandwidth of wireless access networks, a task scheduling problem in edge computing has become the focus of current research. *Márquez-Sánchez et al. (2023)* researched a task scheduling decision-making problem at an application level. They proposed that each application consists of multiple tasks, which should be completed before a deadline. In addition, several researchers carried out task scheduling based on the perspective of edge operations, which cut down resource consumption and improved computational performance. The current work on edge computing task scheduling can be categorized holistically into cross-layer and peer task scheduling.

A cross-layer task scheduling boosts local computing power by handling computation offloading. Distributed computation offloading schemes for goal optimization have

been extensively examined, including energy consumption (*Wen, Zhang & Luo, 2012*; *Chen, 2015*; *Iftikhar et al., 2023*), resolution latency (*Zheng et al., 2019*; *Chen et al., 2016*; *Guo et al., 2019*), bandwidth loss (*Barbera et al., 2013*), and mobility management (*Rahimi, Venkatasubramanian & Vasilakos, 2013*; *Shi, Chen & Xu, 2018*). *Sardellitti, Scutari & Barbarossa (2015)* considered various scenarios comprehensively and designed corresponding algorithms based on user scenarios to optimize computational resources. *Yang et al. (2021)* proposed network-aware computational partitioning for multi-user edge computing. *Li, Ota & Dong (2018)* suggested a joint edge-cloud offloading strategy for deep IoTs. *Kwak et al. (2015)* obtained a task allocation algorithm that optimized CPU and network energy under time delay constraints. *Chen, Zhou & Xu (2018)* suggested an efficient load-scheduling algorithm that can rationalize workload scheduling among edge nodes.

Peer task scheduling extensively conducted by researchers explores the potential of peer cooperation in improving resource efficacy. *Chen, Zhou & Xu (2018)* investigated collaborative cooperation between small base stations (SBS) and designed an online peer-to-peer offloading algorithm to optimize the system's computational performance. *Cui et al. (2017)* implemented centralized scheduling in D2D networks and proposed a novel greedy algorithm for large-scale scheduling problems, thus reducing energy consumption.

Only a few studies have researched cross-layer task scheduling and peer task scheduling. *Champati & Liang (2021)* mainly investigated static task scheduling, while dynamic task scheduling requires many constraints to be realized. The literature does not consider the dynamic scalability of computing resources (*Deng et al., 2016*; *Josilo & Dan, 2019*). In the manuscript, an efficient task scheduling algorithm is provided that can realize efficient task scheduling in complex social relationships.

# CONSTRUCTION OF A SOCIAL RELATIONSHIP-BASED MODEL FOR SCHEDULING EDGE COMPUTING TASKS

## The description of the scenario

In an edge computing environment, mobile devices may experience postpones or poor performance when executing computationally intensive or time-sensitive tasks due to limited device resources. To address this issue, mobile devices can request assistance from nearby devices with available resources, which we refer to as service devices. In the research, we refer to the mobile devices that need scheduling as user equipment or user nodes, while the devices that can assist in task execution are called service devices or service nodes. When multiple user devices send task requests, and there are multiple available service devices, the challenge is to select the appropriate service device to minimize overall delay and energy consumption while ensuring task safety and reliability. To address this challenge, the research combines social relationships between mobile devices to pick service devices for task scheduling to minimize delay and energy consumption during task executions.

A task scheduling scenario in mobile edge computing is considered, where a base station (BS) is equipped with an edge computing server, multiple mobile devices, and multiple service devices. The BS receives real-time task-scheduling requests from the user's

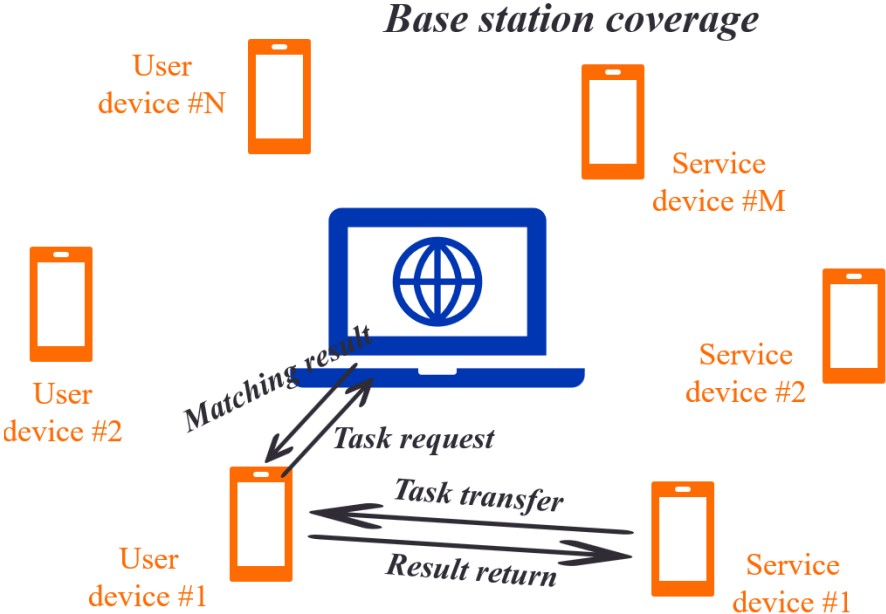

**Figure 1** Schematic diagram of edge computing task scheduling scenario.

equipment and monitors idle service devices. When a user device has a difficult task, it sends a task scheduling request to a BS, which chooses a suitable service device according to a certain strategy and constructs a D2D connection between devices. This allows the user's equipment to transmit a task directly to a selected service device for computation. The task scheduling scenario is illustrated in Fig. 1.

In this scenario, BS monitors mobile devices in real-time. At a certain moment, assume that the user's equipment sends task scheduling requests and that M service devices have idle resources. Each task is represented by a quaternion $<I_i, C_i, O_i, Ttol\ i>$, where $I_i\ C_i, O_i,$ and Ttol indicate the input data for a task, the computing resources required, the output data, the user's maximum tolerance time from sending a task request to its completion, respectively. Tool I reflects a user's sensitivity to delay, as different users have different sensitivity to time. Additionally, each task is indivisible and can only be dispatched to one device. For the number of M service devices, the CPU operating frequency of each device is denoted by fy, reflecting the number of CPU cycles per unit time. Each service device can only assist in executing one task during a round of task allocation. The edge server uses the KM algorithm to pick a service device for task execution based on task requirements, computing resources, social relationships, and other information. The matching result is sent back to the user equipment, which executes the task locally or sends a task scheduling request to the service device matched by the BS *via* a D2D connection. Eventually, the outcomes are attained by the local execution or executed by the service device and transmitted to the user equipment.

## A scheduling method

In the task scheduling method of edge computing based on social relationships, the following steps are taken to schedule tasks:

A Construction of User Social Relationship Graph: Leveraging social assessment data within a user social evaluation dataset, social relationships among user devices and service devices are analyzed and quantified. A comprehensive user social relationship graph is meticulously constructed by implementing mathematical convolution and the Jaccard similarity ratio. Also, task execution costs are computed based on task requirements and device information.

The Construction of Edge Computing Task Scheduling Model: A task-device bipartite graph and task scheduling model are combined to construct a model used for edge computing task scheduling. The model considers factors such as task execution delay, the energy consumption of a task execution, and the energy consumption of a user equipment standby.

KM Algorithm-based Solution: The edge computing task scheduling model is resolved by implementing the KM algorithm, presenting the result of a task scheduling that minimizes delay and energy consumption.

The scheduling method ensures that tasks are scheduled to be executed on safe and reliable service devices by analyzing and quantifying a social relationship between devices. The physical distance between a user and service devices is also considered to construct an energy-efficient D2D connection. The method ultimately provides a task scheduling result that minimizes delay and energy consumption. The task scheduling method based on social relationships in edge computing is illustrated in Fig. 2.

## Building a social relationship graph

In D2D communication, task offloading requires a user node to transmit a task to a service node *via* a D2D connection for processing. However, malicious service nodes in a network can disrupt node cooperation, compromising user privacy and causing incomplete task execution. To ensure safe task offloading, social trust between a user and a service device holder is examined and quantified through social relationships between mobile devices. A social trust relationship enables users to discern secure and dependable service devices for efficient task offloading. Moreover, given the prevalence of altruistic behaviors within individuals maintaining social ties, the inclination of service device holders to share their resources can be assessed through an analysis of social relationships among individuals with mobile devices. Thus, the research considers social relationships between users as a crucial basis for designing task scheduling strategies and represents social relationships between users through the graph of a user's social relationship.

A social relationship graph $G_s$ between users is represented by a (N+M) by (N+M) matrix ($P_{ij}$), storing social relationship scores, which are uniformly maintained and updated by the edge server of BS. After a service node provides computing services for a user node, a user node can rate the service node, and vice versa. The rating is an integer ranging from 0 to 5, with higher scores indicating greater satisfaction with the other party's performance in the cooperation, like rating scheme used for food delivery service or taxi
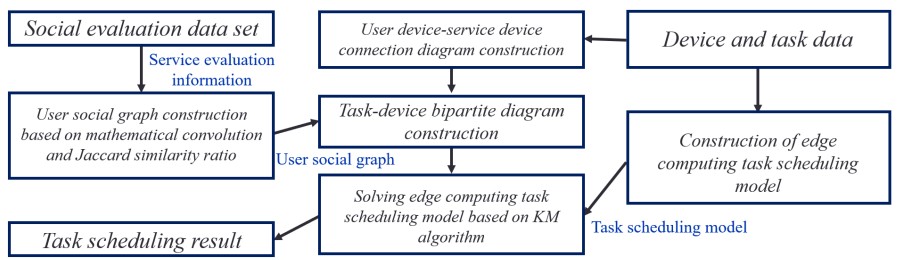

**Figure 2** Edge computing task scheduling method based on social relationships.

service providers in everyday life, service providers can also rate their cooperation with users. Thus, a service node can also rate a user node, and an assessment between the 2 is mutual but not necessarily equal.

The research introduces the computation of the social relationship score of service nodes from the perspective of user nodes. A social relationship score in the social relationship graph can be found based on the service assessment factor, recommendation trust factor, and similar recommendation factor. The calculation methods for these 3 decision factors are given as follows:

### Social relationship computing

(1) Service assessment factors

The social relationship graph can store a social relationship score between user node i and service node j if they have had a direct interaction. This score is calculated by employing the service assessment of node i to node j and stored. Assuming the latest H service evaluation records of node i to node j are $S_{ij} = \{S_{ij}(1), S_{ij}(2), \ldots, S_{ij}(k), \ldots, S_{ij}(H)\}$, where $S_{ij}(k)$ represents scores given by user node i to service node j, $k \in [1, H]$. H represents the number of service assessments of user node i to service node j within a specific period to ensure the effectiveness of service assessment. The research employs the mathematical convolution concept (*Wei, Yang & Liu, 2023*) to derive Eq. (1) to compute the service assessment factor of user node i to service node j. To facilitate the processing of the proportion of each decision factor in the social relationship, service assessment factor scores are normalized by implementing Eq. (2).

$$Q_{ij} = \frac{1}{H} \sum_{k=1}^{H} S_{ij}(k) \times \frac{1}{H-k+1} \tag{1}$$

$$Q_{ij} = \frac{Q'_{ij} - \min(Q_{ij})}{\max(Q_{ij}) - \min(Q_{ij})} = \frac{Q'_{ij}}{5} \tag{2}$$

where $Q_{ij}'$ represents the service assessment calculated directly through historical evaluation. After normalization is run, $Q_{ij}$ represents the service assessment factor, with $\max(Q_{ij})$ and $\min(Q_{ij})$, which are the maximum and minimum scores of a service evaluation, 0 and 5, respectively.

**(2) Recommended trust factor**

When there is no direct assessment between a user node and a device node, a user node can still calculate the social relationship score by referring to other nodes' assessment of the service node since trust can be transferred. An intermediary node is a node assessed by user nodes and has assessments for service nodes.

Let the intermediary node set of user node i and service node j be $V_{ij}$, $V_{ij} = \{V_{ij}(1), V_{ij}(2), \ldots, V_{ij}(k), \ldots, V_{ij}(|V_{ij}|)\}$, where $|V_{ij}|$ represents the number of intermediary nodes. The recommendation trust $W_{ij}$ of the user node i to the device node j can be computed based on the service assessment of user node i to the intermediary node by employing Eq. (3).

$$W_{ij} = \frac{1}{|V_{ij}|} \sum_{k=1}^{|V_{ij}|} Q_{ik} \times Q_{kj} \tag{3}$$

**(3) Similar recommendation factor**

When the user node i has not assessed the service node j, the user node i can also refer to the evaluation of the service node j by nodes with similar assessments to itself to judge the credibility of the nodes (*Park & Tussyadiah, 2020*). Assuming that node $K_{ij}(\eta)$ has not only assessed the same node as user node i , but also assessed node j, then $K_{ij}(\eta)$ represents a reference node. The set of reference nodes is denoted by $K_{ij} = \{ K_{ij}(1), K_{ij}(2), \ldots, K_{ij}(\eta), \ldots, K_{ij}(|K_{ij}|)\}$ and the size of the set is denoted by $|K_{ij}|$. $N_\xi$ denotes the intersection set of nodes serving user node i and reference node $\eta$, and the set size is represented by $|N_\xi|$. The research employs the generalized Jaccard similarity coefficient to attain the similarity score between user node i and reference node $\eta$ on the same group of service nodes $N_\xi$, as shown in Eq. (4).

$$J_{in} = \frac{\sum_{\xi=1}^{|N_\xi|} \min\{Q_{i\xi}, Q_{\eta\xi}\}}{\sum_{\xi=1}^{|N_\xi|} \max\{Q_{i\xi}, Q_{\eta\xi}\}} \tag{4}$$

**(4) Calculation of social relationship score**

Users tend to trust their judgments more than others' opinions. Therefore, when there is a direct assessment between user node i and service node j, the social relationship between them in the social relationship graph is determined by the service assessment, represented by $P_{ij} = Q_{ij}$.

However, when there is no direct assessment between user node i and service node j, the social relationship must be computed based on the recommendation trust factor $W_{ij}$ and the similar recommendation factor $A_{ij}$. The impact of $W_{ij}$ and $A_{ij}$ on the social relationship $P_{ij}$ can be gauged by the number of intermediary nodes $|V_{ij}|$ and the number of reference nodes $|K_{ij}|$.

The social relationship score $P_{ij}$ of user node i to service node j in a user social relationship matrix P can be obtained by implementing Eq. (5):

$$P_{ij} = \begin{cases} Q_{ij}, & H \neq 0 \\ \varphi \times W_{ij} + (1-\varphi) \times A_{ij}, & H = 0, |V_{ij}| + |K_{ij}| \neq 0. \\ 0.5, & |V_{ij}| + |K_{ij}| = 0 \end{cases} \tag{5}$$

### Construction of social relationship graph

The process of constructing and implementing a user social relationship graph Gs is presented as follows: Firstly, the number of N user nodes and the number of M service nodes are obtained. Next, service rating information between mobile devices is obtained. Then, for each user node, a social relationship score between it and each service node is computed.

To determine whether to add an edge between a user node and a service node in the social relationship graph, the social relationship score P between them is compared to a set threshold $\delta$. An edge is added if P is greater than or equal to $\delta$. Otherwise, the social relationship graph is not modified, and the next service device is monitored.

## The construction of a connection graph

Given that the maximum communication distance of D2D is represented by $L_{max}$, the location coordinates of the user equipment i are (x1 i, y1 i), and the location coordinates of service equipment are (x2 j, y2 j), a connection diagram between the user equipment and service equipment is denoted as Gd. Equation (6), the Euclidean distance formula, is used to compute the distance between user equipment i and service equipment j within the BS coverage.

$$L_{ij} = \sqrt{(x_i^1 - x_j^2)^2 + (y_i^1 - y_j^2)^2}. \tag{6}$$

The connection graph between user equipment and service equipment, Gd, can be constructed by attaining the serial numbers of N user nodes, the serial numbers of M service nodes, and their respective location information. The physical distance, L, between each user and service equipment, can then be computed. If L is less than or equal to the maximum D2D communication distance, $L_{max}$, a link between the user and service equipment in the Gd edge, can be added. On the other hand, if the condition is not met, the device connection diagram remains unmodified.

## The construction of bipartite graph

To optimize the overall scenario by minimizing delay and energy consumption, the task-device bipartite graph G is constructed by combining the user equipment-service equipment connection graph Ga and the user social relationship graph Gs of mobile devices.

Figure 3 illustrates the task-device bipartite graph, consisting of 2 disjoint subsets, X and Y. The set X represents the tasks that users must perform. In contrast, set Y represents the mobile devices capable of performing tasks. The set Y comprises user and service equipment, represented in solid circles in Fig. 3.

In any case, a user can execute a task locally, resulting in an edge relationship between the user's task i and their equipment in the task-device bipartite graph. However, due to limited resources, users may seek task offloading and thus cannot provide computing services for other user's tasks. Therefore, there are no associated edges between user task i and other user devices, meaning user devices cannot serve other devices.

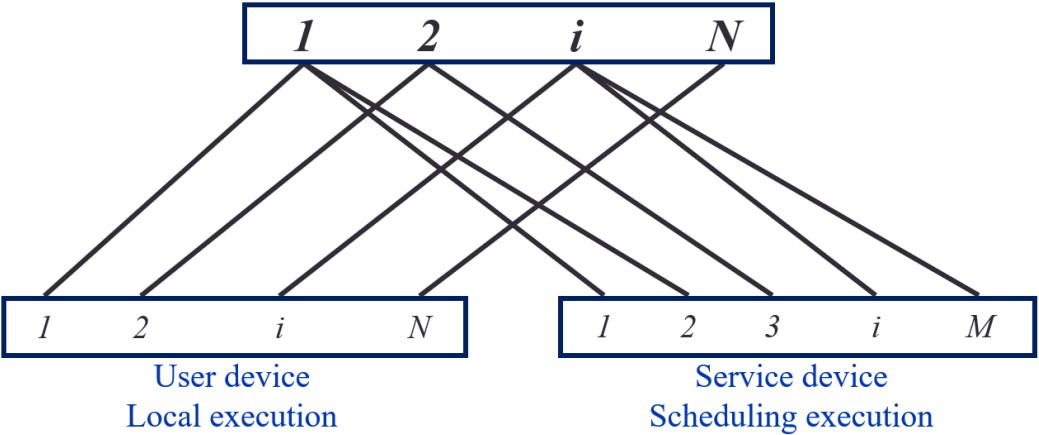

**Figure 3** Diagram of the task-equipment bipartite.

## The construction of edge computing task scheduling model
### Calculation of communication cost

Under the control of the BS, any user equipment can construct a D2D connection with a serving device. Since tasks are typically time-sensitive, the BS only constructs and maintains one D2D connection per mobile device. A D2D communication is adopted among mobile devices, assuming orthogonal spectrum resources and no interference between devices. According to Shannon's theorem, a D2D communication transmission rate between user equipment i and service equipment j can be calculated by Eq. (7):

$$R_{ij} = B\log_2\left(\frac{P_i^t|D_{ij}|^2 L_{ij}^{-\gamma}}{\Sigma_{U_{i'}\in U, i'\neq i} P_i|D_{ij}|^2 L_{ij}^{-\gamma} + \sigma^2} + 1\right) \tag{7}$$

where B, PT i, and $D_{ij}$ represent the available bandwidth of the transmission channel, the transmission power of the user equipment i, and the channel parameter of the D2D transmission. The path loss between user device i and serving device j is L $-\gamma$ ij, where L, $\gamma$, $\sigma 2$ represent the physical distance between mobile devices, the path loss exponent, and the noise variance.

In a task transmission, the mobile device can only construct one D2D connection at a time, and the mobile device, as the server, has relatively limited computing power and can only receive one task to do a calculation at a time. The time it takes for user equipment i to transmit a task to service equipment j is the data transmission delay Td ij. The transmission delay caused by task scheduling of user equipment to service equipment can be computed. When tasks are scheduled to service nodes for calculation, two parts of energy consumption will be generated. One is the energy consumption caused by user equipment i scheduling tasks to service equipment j, and the other is the energy consumption led by service equipment j sending task execution outcomes back to user equipment i. Total energy consumption, the sum of the energy consumption of these 2 parts, constitutes the energy consumption Ed ij of scheduling transmission. Equations (8) and (9) are used to do calculations as follows:

$$T_{ij}^d = \frac{I_i}{R_{ij}} + \frac{O_i}{R_{ji}} \tag{8}$$

$$E_{ij}^d = \frac{I_i}{R_{ij}} P_i^t + \frac{O_i}{R_{ji}} P_j^t. \tag{9}$$

### The cost calculation of a task execution

User devices have two options for task execution: local execution or scheduling tasks on other mobile devices. When a task is executed locally on a user device, the cost of task execution includes the energy consumption and computational delay produced by the device itself. On the other hand, when a user device schedules a task to be executed on a service node, the cost of the energy consumption of task execution includes the energy consumption for task transmission, the energy consumption for computing tasks on the service device, the standby energy consumption of the user device waiting for the task to return, and the delay cost, which includes task transmission and task computation delays.

To execute locally, Eqs. (10) through (16) are used:

$$T_i^{loc} = \frac{I_i C_i}{F_i} \tag{10}$$

$$E_i^{loc} = P_i^c T_i^{loc}. \tag{11}$$

To execute scheduling, the relevant equations are given as follows:

$$T_{ij}^c = \frac{I_i C_i}{F_j} \tag{12}$$

$$E_{ij}^c = P_i^c T_{ij}^c \tag{13}$$

$$T_{ij}^{sch} = T_{ij}^d + T_{ij}^c \tag{14}$$

$$E_i^w = T_{ij}^{sch} \times E_o \tag{15}$$

$$E_{ij}^{sch} = E_{ij}^d + E_{ij}^c + E_i^w. \tag{16}$$

### Task scheduling model

Let $\lambda_{ij}$ set as a binary decision variable of task scheduling. If a task of a user node i is offloaded to the service node j, then $\lambda_{ij} = 1$; otherwise, $\lambda_{ij} = 0$. It means the task is not executed on the service node. The task-device bipartite graph is represented by G = {N, $\varepsilon$}, where eij represents the associated edge between user task i and device node j, which satisfies the social and communication conditions required for task scheduling. To sum up, this research aims to minimize the execution delay and energy consumption of all tasks in the entire scenario, and the objective function is shown in Eq. (17):

$$\min_{\lambda} \sum_{i=1}^{N} \left[ \left(1 - \sum_{j=1}^{M} \lambda_{i,j}\right) Z_i^{loc} + \sum_{j=1}^{M} \lambda_{i,j} Z_{ij}^{sch} \right]$$

$$= \min_{\lambda} \frac{1}{2} \sum_{i=1}^{N} [(1 - \sum_{j=1}^{M} \lambda_{i,j})(T_i^{loc} + E_i^{loc}) + \sum_{j=1}^{M} \lambda_{i,j}(T_{ij}^{sch} + E_{ij}^{sch})]. \tag{17}$$

# SOLUTION TO EDGE COMPUTING TASK SCHEDULING MODEL USING KM ALGORITHM

## Construction and implementation of user social relationship graph

The edge server of a BS obtains service assessment information and computes the social relationship score between user equipment and service equipment implementing the formula from the previous analysis. It then determines whether the social relationship score meets the social threshold score to construct the user's social relationship graph. The user's social relationship graph is implemented in Java and stored as an array. When there is no social relationship between the user and the service equipment, the score is set to 0. When there is a social relationship, the score is assigned to 1. Since user equipment and service equipment are disjoint sets with no associated edges within the same set, the user's social relationship graph represents a bipartite graph. Although we do not need to consider the bipartite graph property of the social relationship graph in this section, it facilitates the construction of the task-equipment bipartite graph. The flow chart showing the construction of the user's social relationship graph is presented in Algorithm 1.

## Construction and implementation of user equipment-service equipment connection graph

The mobile device's location information is attained to compute the physical distance between the user equipment and service equipment and determine whether a D2D communication condition is met to construct the user equipment-service equipment connection graph. The user equipment-service equipment connection graph is implemented in Java and stored as an array. When a D2D communication condition is not satisfied, the score is set to 0, and when it is satisfied, the score is assigned to 1. The user equipment-service equipment connection graph is also bipartite since user equipment and service equipment are disjoint sets, with no associated edges within the same set. We do not need to consider the bipartite graph properties of the user equipment-service equipment connection graph here.

## Construction and implementation of task-equipment bipartite graph

A task-equipment graph is constructed by implementing the relationship between the user's social relationship graph and the edges in the user equipment-service equipment connection graph. The task-equipment bipartite graph is implemented in Java and stored as an array. When a social relationship between user equipment and service equipment and D2D communication conditions are met, the array score is assigned to 1. If no social relationships or D2D communication conditions are not met, the array score is assigned to 0. If there is no social relationship and D2D communication conditions are not met, the array score is also set to 0.

**Table 1    The implementation process of user social relationship graph construction algorithm.**

| Algorithm 1: User social relationship graph construction algorithm |
| --- |
| **Input:** Number of user devices N, number of server devices M, service evaluation information, threshold P |
| **Output:** Social Relationships Pij between user device i and service device j |
| **Begin** |
| 1:     **if** Pij>P |
| 2:    Social[i][j]=1 |
| 3:    j+1 |
| 4:     **else if** Pij<=P  **then** |
| 5:    j+1 |
| 6:      **if** j>M |
| 7::    i+1 |
| 8:        **if** i>N |
| 9:          i+1 |
| 10:        **else if** |
| 11:     **else if** j<=M  **then** |
| 12:       return Pij |
| 13:     **End if** |
| 14:    **End** |

## Solution algorithm and implementation of edge computing task scheduling model based on KM algorithm

Based on a task-equipment bipartite graph, the edge computing task scheduling model is constructed to attain the weights of the edges in the bipartite graph, which can be resolved by implementing the KM algorithm.

According to the proposed model, the specific execution steps for resolving the edge computing task scheduling model based on the KM algorithm are as follows:

Step 1: Compute the task execution cost as the weight of the bipartite graph edge.

Step 2: Set the top mark for the task and device nodes.

Step 3: Construct equal subgraphs.

Step 4: Find augmenting paths in equal subgraphs. If an augmenting path is found, go to Step 5. If not, go to Step 6.

Step 5: Determine whether a complete match is attained. If not, return to Step 4. If so, end.

Step 6: Compute the top mark's modified score, modify it, and return to Step 3.

## EXPERIMENTAL ANALYSIS

The experimental execution environment for the research is macOS 10.13.6 with a 1.4 GHz Intel Core i5 processor and 4GB of memory. Dataset 5 was collected from a taxi-hailing service program in 2022, and the research implements passenger and driver information from the dataset. The proposed model is compared with three other algorithms: Pure KM(PKM), which is the KM without social relations; improved Genetic Algorithm (IGA) (*Zhou et al., 2018*), which aims to reduce delay and energy consumption; and JPORA (*Saleem et al., 2020*), which aims to decrease overall task execution delay.

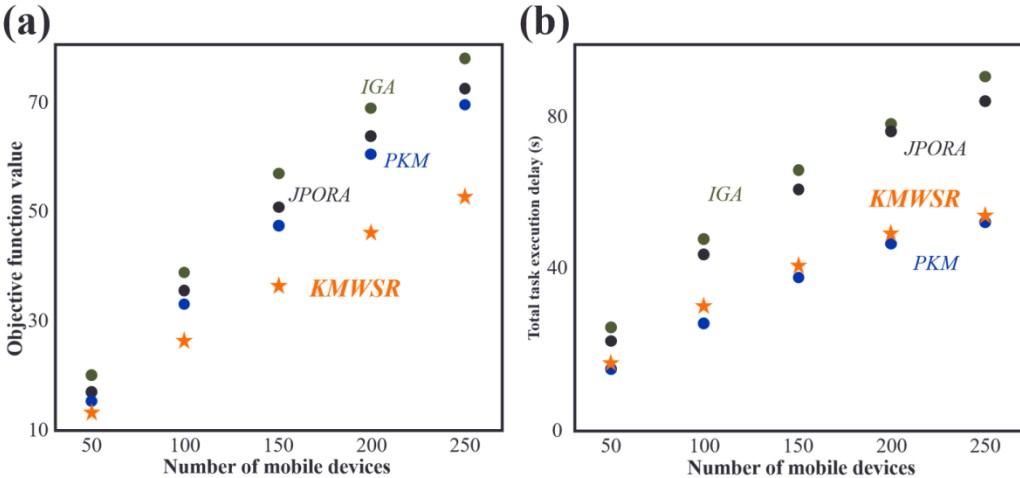

**Figure 4** Comparisons of (A) objective function values and (B) task execution delay when the number of devices changes.

## Comparisons of objective function scores and task execution delay when the number of devices changes

As shown in Fig. 4A, the objective function score increases as the number of mobile devices rises. The increase in the number of devices leads to a rise in the tasks of user equipment, resulting in increased delay and energy consumption. The proposed KMWSR performs the best, followed by the JPORA, with the PKM and IGA ranked third and fourth, respectively. The proposed algorithm considers both task execution delay and energy consumption and the social relationship between mobile devices, allowing user devices to be offloaded to trusted service devices, resulting in better task execution and a lower probability of task rejection. The JPORA reduces task execution delay, the PKM attains the optimal match, and the IGA attains an approximate optimal solution. The objective function score's growth rate slows as the number of mobile devices rises due to the increasing density of service devices that user equipment can choose from.

After reviewing Fig. 4B, the JPORA prioritizes reducing task execution delay and outperforms the KMWSR in this aspect. Although the time delay curves of the PKM and IGA are similar, the PKM achieves better results since it acquires the optimal solution for matching user tasks with service devices. In contrast, the IGA produces an approximate optimal solution. The total latency of tasks performed by user devices increases more slowly as the number of mobile devices grows due to the unchanging mobile range and maximum communication distance of D2D communication. This increased density raises the number of service devices with good performance available around the user device, making it simpler to schedule tasks on mobile devices closer in proximity. However, the total latency of the JPORA, PKM, and IGA increases more rapidly than the KMWSR because these three algorithms do not consider social relationships among device users. As a result, service devices may reject some tasks, and only local executions in areas with

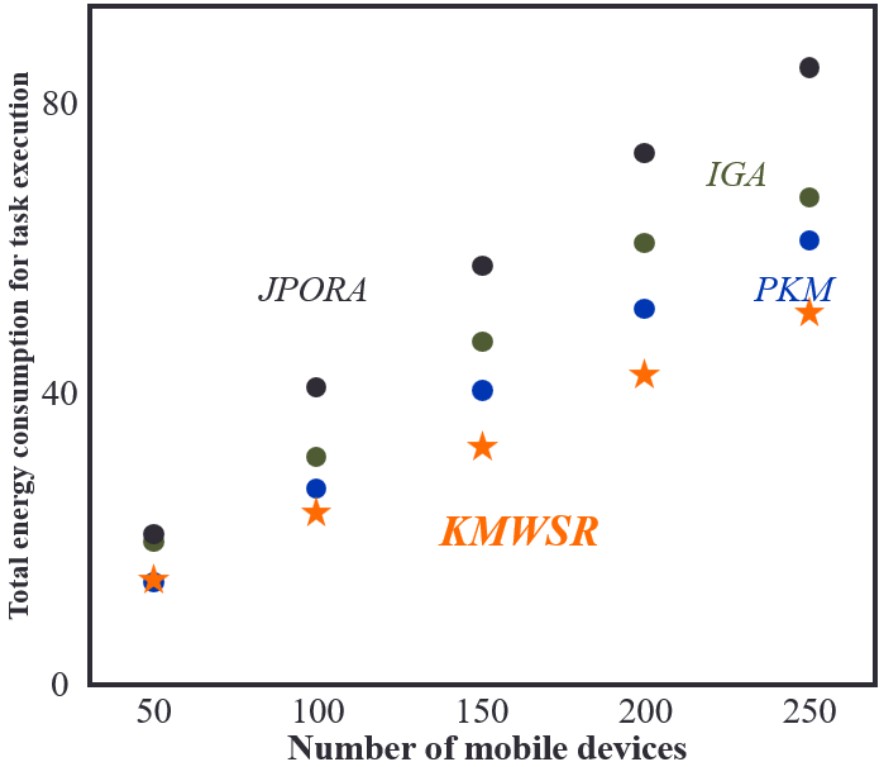

**Figure 5**  Comparisons of energy consumption when the number of devices changes.

relatively scarce computing resources are feasible, leading to a faster total delay growth as shown in Fig. 4.

## Comparisons of energy consumption when the number of devices changes

Figure 5 shows that as the range of mobile devices remains constant, the total energy consumption of task execution increases with the number of mobile devices. Among the algorithms assessed in the research, the proposed KMWSR is the most effective in terms of energy saving, followed by the PKM. The IGA attains an approximate optimal solution and has a slightly lower energy-saving effect than the PKM. On the other hand, the JPORA, focusing more on decreasing time delay, has the worst performance in terms of energy consumption. This is because the JPORA does not consider energy consumption as much as the other algorithms. Eventually, the proposed KMWSR achieves relatively the best energy consumption even though the number of mobile devices increases.

## Comparison of objective function scores when social relationship threshold changes

Figure 6 illustrates the change in the objective function as the social relationship threshold increases while the number of mobile devices remains constant. It shows that when the social relationship score increases, the objective function score attained by all four

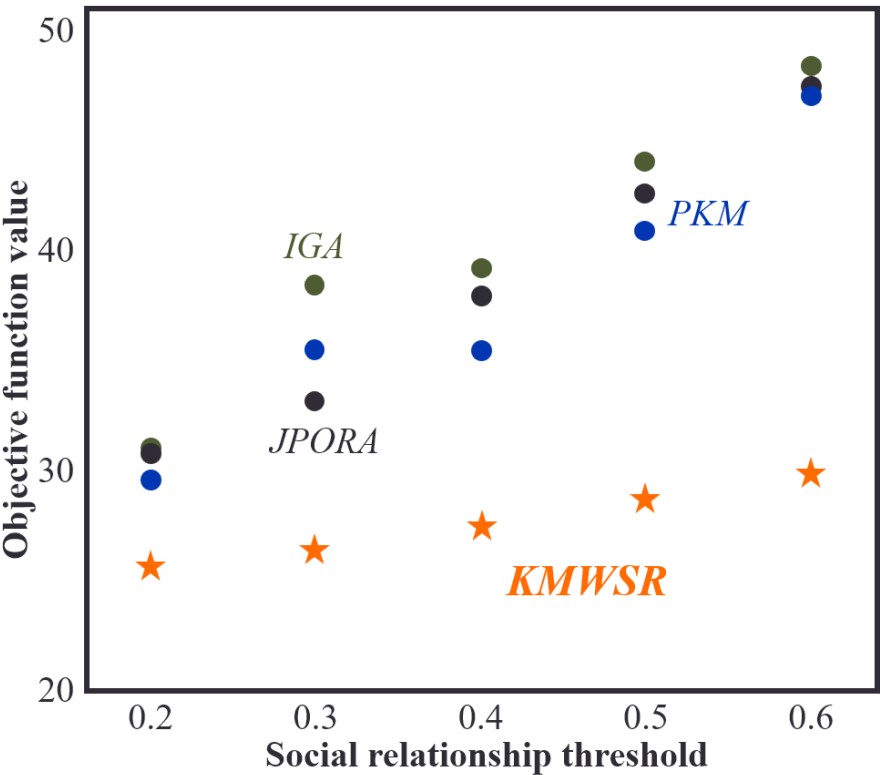

**Figure 6** Comparison of objective function values when social relationship threshold changes.

algorithms also increases. This is because a higher social relationship threshold increases the likelihood of task rejection by service devices and dispatching to devices with average performance or local execution, resulting in increased delay and energy consumption. The KMWSR performs the best in this scenario, with only a slight increase in the objective function score. The KMWSR considers social relationships and excludes devices with weak connections, preventing task rejection. However, increasing the social relationship score reduces the number of devices in the candidate service set, potentially excluding devices with good performance, leading to an increase in the objective function score.

**Comparison of task execution delay and energy consumption of task execution when the number of devices and the social relationship threshold change**

Figure 7A shows that when the number of mobile devices remains constant, the task execution delay of the four algorithms increases with the rise of the social relationship threshold score. This is because the number of service devices remains the same, and as the social threshold increases, users with weak social connections find it difficult to find service devices willing to accept tasks. Among the algorithms assessed in the research, the proposed KMWSR is the least affected by the change in social threshold score, and its performance is the most stable. This is because the KMWSR considers the social relationship between

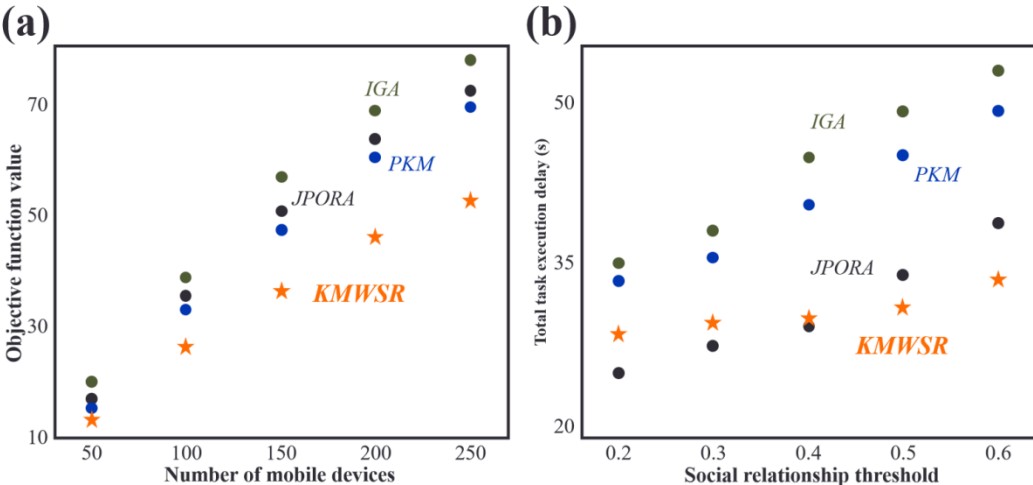

**Figure 7** Comparison of (A) task execution delay when the number of devices changes and (B) energy consumption of task execution when the social relationship threshold changes.

devices and excludes service devices unwilling to accept tasks in advance, thus preventing the possibility of service devices matching edge servers and rejecting tasks. Consequently, the tasks of user devices will be scheduled to the optimal available server.

Figure 7B illustrates that as the social relationship score increases, the energy consumption of task execution also rises for all four algorithms when the number of user devices remains constant. The proposed KMWSR is the least affected by changes in the social relationship score and performs the best in reducing energy consumption. This is because it considers the social relationship between users and avoids task rejections. However, even the KMWSR is slightly affected by increased social threshold as it reduces the number of available service devices, potentially excluding devices with better performance.

When a social relationship score is low, the PKM and IGA exhibit similar outcomes to that of the KMWSR. Nevertheless, with an elevation in the social threshold, the effectiveness of these two algorithms experiences a notable decline. This phenomenon stems from the heightened social relationship score, increasing the probability of service device task rejections. Consequently, user devices must opt for suboptimal servers or perform local task execution, amplifying energy consumption. Notably, the JPORA, prioritizing delay reduction, demonstrates inferior energy-saving performance.

## The comparison of task execution delay when the number of task changes

Figure 8 presents the delay time of task execution of distinct algorithms as the social relationship threshold grows for distinct numbers of tasks in non-specific cases. The delay time of the four algorithms in executing the task grows as the social relationship threshold increases.

(a) The number of tasks is 100
(b) The number of tasks is 200

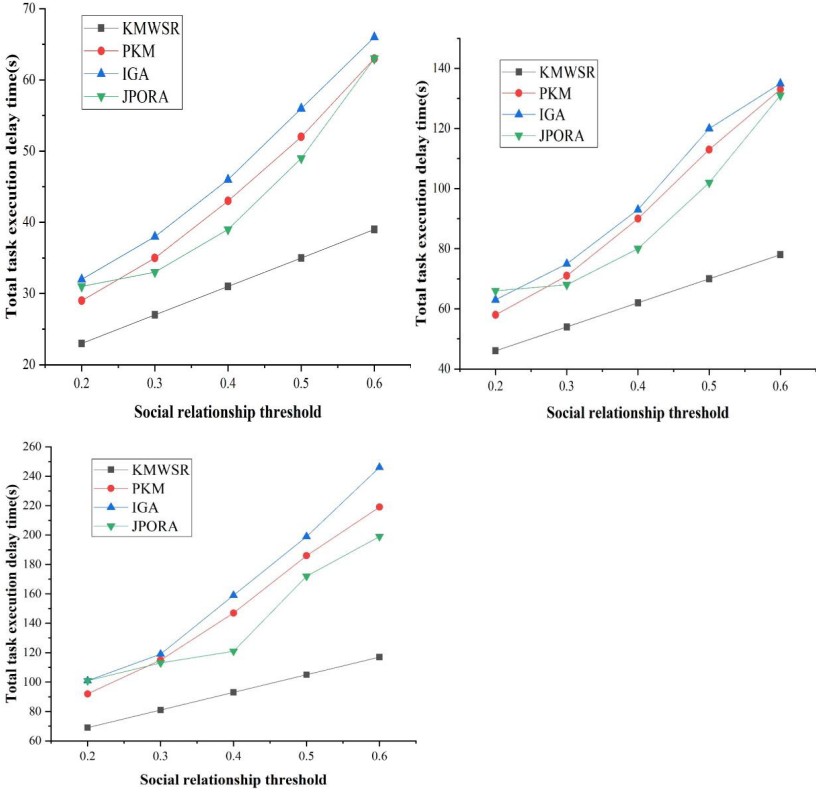

**Figure 8** **Comparison of task execution delay when the number of tasks changes.**

(c) The number of tasks is 300

The proposed KMWSR can ensure the stable growth of task execution delay under distinct numbers of tasks since the KMWSR can ensure the allocation of computational resources under the premise of considering the social relationship, thus realizing the reasonable scheduling of tasks. The remaining three algorithms exhibit a substantial and escalating growth pattern as the task count rises. This trend suggests challenges in efficiently allocating computational resources through social relations division when confronted with varying task volumes, leading to diminished scheduling performance. The JPORA performs like the proposed algorithms. However, as task count rises, there is an elevated likelihood of task rejection for execution by the JPORA, resulting in a performance decline. This phenomenon is also observed in the PKM and the IGA, where the performance of the same algorithm is adversely affected by an increased task count. Similarly, the performances of the PKM and IGA are also severely affected by the number of tasks, and the task execution delay increases significantly. This shows that the proposed KMWSR is more effective for scheduling massive tasks.

In summary, alterations in the social relationship threshold score exert a discernible impact on task execution delay and energy consumption. The proposed algorithm exhibits a relatively lower susceptibility to such variations, showcasing superior performance when compared to the other three algorithms within the social relationship score range of

0.2 to 0.4. This range balances energy conservation and task load reliability, facilitating efficient and stable scheduling for substantial task volumes. In practical scenarios integrated with the BS, the edge server should adjust the social relationship threshold dynamically based on real-time user social relationships. This adaptive approach ensures the provision of high-response, low-energy task scheduling services while upholding the safety and reliability of task executions.

## CONCLUSION

The research introduces an innovative edge computing task scheduling model grounded in user social relations, resolved through the KM. The contribution encompasses a task scheduling approach for edge computing, intending to minimize latency and energy consumption while ensuring secure and dependable task scheduling. Service evaluation, recommendation trust, and analogous recommendation factors are combined to quantitatively assess social relationships among users. Constructing a user social relationship graph based on these scores guarantees the safety and reliability of task scheduling. To facilitate D2D communication connections, the physical distance is computed between mobile devices by implementing Euclidean distance, forming a user device-service device connection graph. The synthesis of the user social relation graph and the user device-service device connection graph results in a task-device bipartite graph. An edge computing task scheduling model is formulated that accounts for both the local execution cost and the scheduling execution cost of tasks. Ultimately, the KM is applied to determine the model and validate its efficacy through comparisons with mainstream algorithms. The research findings indicate that leveraging D2D communication technology for task transmission mitigates delay, energy consumption, and core network communication pressure.

### Funding
The authors received no funding for this work.

### Competing Interests
The authors declare there are no competing interests.

### Author Contributions
- Yun Teng conceived and designed the experiments, performed the experiments, analyzed the data, performed the computation work, prepared figures and/or tables, authored or reviewed drafts of the article, and approved the final draft.
- Zijia Liu conceived and designed the experiments, performed the experiments, analyzed the data, performed the computation work, prepared figures and/or tables, authored or reviewed drafts of the article, and approved the final draft.

### Data Availability
The code is available in the Supplemental File.
The dataset is available at: https://tianchi.aliyun.com/dataset/76359.

## Supplemental Information

Supplemental information for this article can be found online at http://dx.doi.org/10.7717/peerj-cs.2457#supplemental-information.

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
