# Peer review of "Edge computing task scheduling method based on user’s social relations: a construction and solution for Smart City Library"

_PeerJ Computer Science, doi:10.7717/peerj-cs.2457_

## Round 0.1 · original submission · Major Revisions

Thank you for submitting your manuscript, After careful evaluation by our review team, we appreciate the effort and depth of research that has gone into your work. However, please take into consideration the reviewer's comments and mine below
Please carefully review the attached comments from the reviewers and make the necessary revisions. We look forward to receiving your revised manuscript.

Thank you for your understanding and cooperation.

Editor Comments: please provide the reason for selecting KM and demonstrating its advantages over other methods in this context .

Including quantitative data, graphs, and comparisons with baseline models or existing techniques would give readers a clearer picture of the proposed model's effectiveness.

Please carefully improve the paper language

Reviewer 1 ·

Basic reporting

Some figures, such as the energy consumption comparison (Figure 5), could benefit from additional explanations in the caption to improve reader understanding.

Provide more details on the construction process of the user social relationship graph for a better understanding of the methodology.

Include clearer diagrams and figures to illustrate complex models and algorithms.

Include a section on the robustness of the proposed method to adversarial attacks or malicious behavior in edge computing environments.

The conclusion could be strengthened by mentioning specific future directions, such as incorporating machine learning techniques for better social relationship modeling or addressing network dynamism.

There is no mention of limitations. It would be helpful to include a brief discussion of the approach's potential drawbacks.

Experimental design

The choice of KM algorithm is justified, but there’s no comparison with alternative algorithms (e.g., genetic algorithms, particle swarm optimization) that might also suit this problem.

Explain in detail how Euclidean distance is utilized in crafting the user equipment-service equipment connection graph.

Include a comparison with existing edge computing task scheduling methods to highlight the novelty of the proposed approach.

Provide a more comprehensive literature review on edge computing, task scheduling, and social relationship-based models.

Consider discussing the long-term sustainability and maintenance requirements of implementing the proposed task scheduling model.

Explain how interference is managed during D2D communication to ensure reliable task scheduling.

Validity of the findings

The experimental setup lacks sufficient detail. For example, how were social relationships simulated, and what metrics were used to evaluate their strength?

The dataset from a taxi-hailing service is used, but the relevance of this dataset to Smart City Libraries is not fully explained. A more domain-specific dataset would make the experiments more convincing.

Clarify the specific metrics used in quantifying user social relationships through mathematical convolution and the Jaccard similarity ratio.

Discuss the potential implications of network congestion and latency on the performance of the scheduling model.

Discuss potential challenges in implementing the model in a real-world smart city library setting.

Explain how the graph is updated over time with new data to maintain accuracy and relevance.

Provide criteria for selecting service devices beyond proximity and resource availability, such as reliability.

Evaluate the model’s flexibility in adapting to different environments and scenarios, highlighting its adaptability.

Consider implementing a feedback loop for continuous improvement of the model based on real-world usage.

Reviewer 2 ·

Basic reporting

1. The abstract could be enhanced by adding specific numerical results or performance improvements to substantiate the claim of "positive impacts."
2. In abstract the term "positive impacts" is too vague. It would be helpful to specify in what terms these impacts are measured (e.g., delay reduction, energy saving, etc.).
3. The introduction could benefit from a clearer definition of “Smart City Libraries” and why edge computing is particularly suited to address their task scheduling challenges.
4. The review of related works covers a broad range of approaches to task scheduling. However, the discussion is a bit too high-level and lacks a critical assessment of the limitations of previous models.
5. The transition between peer-to-peer task scheduling and cross-layer task scheduling should be smoother. It feels abrupt and could confuse readers.
6. References to previous studies are listed, but it is not clear how the proposed method differs or improves upon them. This gap should be addressed to emphasize the originality of the work.
7. While the Euclidean distance for device-to-device connections is reasonable, it might oversimplify the task scheduling model. Adding considerations for network congestion or bandwidth variability could provide a more realistic model.
8. The description of the KM algorithm is somewhat superficial. A deeper explanation of how the algorithm is tailored for this specific edge computing task scheduling context is necessary.
9. There is a lack of discussion on the computational complexity of the proposed scheduling method. Since edge devices are resource-constrained, an analysis of how efficient the scheduling method is would be useful.
10. The method does not seem to address fault tolerance. What happens if a task cannot be completed due to device failure or unreliable social relationships?

Experimental design

See "Basic reporting".

Validity of the findings

See "Basic reporting".

Additional comments

No comments.

---

## Round 0.2 · accepted · Accept

Dear authors

We have now received feedback from the experts on your revised manuscript. Bases on their input, I'm pleased to inform you that your manuscript now qualify for the acceptance criteria.

Congratulations and thank you for your valuable contribution

Reviewer 1 ·

Basic reporting

The revised version is clear, unambiguous, technically correct.
The article include sufficient introduction and background.
Relevant literature is appropriately referenced.

Experimental design

Research scope and objectives are clear.

Validity of the findings

Rigorous investigation performed by authors.
Conclusions is well stated.

Reviewer 2 ·

Basic reporting

The authors have successfully addressed all the issues and the paper is ready to be accepted.

Experimental design

The authors have successfully addressed all the issues and the paper is ready to be accepted.

Validity of the findings

The authors have successfully addressed all the issues and the paper is ready to be accepted.